# Noradrenaline Sensitivity Is Severely Impaired in Immortalized Adipose-Derived Mesenchymal Stem Cell Line

**DOI:** 10.3390/ijms19123712

**Published:** 2018-11-22

**Authors:** Pyotr A. Tyurin-Kuzmin, Vadim I. Chechekhin, Anastasiya M. Ivanova, Daniyar T. Dyikanov, Veronika Y. Sysoeva, Natalia I. Kalinina, Vsevolod A. Tkachuk

**Affiliations:** 1Department of Biochemistry and Molecular Biology, Faculty of Medicine, Lomonosov Moscow State University, 119991 Moscow, Russia; v-chech@mail.ru (V.I.C.); ivanovanastasia14@gmail.com (A.M.I.); danidy@inbox.ru (D.T.D.); veroniks@mail.ru (V.Y.S.); tkachuk@fbm.msu.ru (V.A.T.); 2Laboratory of Molecular Endocrinology, Russian Cardiology Research Center, 121552 Moscow, Russia

**Keywords:** mesenchymal stem/stromal cells, noradrenaline, norepinephrine, intracellular signaling, membrane receptors, adrenergic receptors, calcium imaging, ASC52telo, hTERT

## Abstract

Primary adipose tissue-derived multipotent stem/stromal cells (adMSCs) demonstrate unusual signaling regulatory mechanisms, i.e., increased of sensitivity to catecholamines in response to noradrenaline. This phenomenon is called “heterologous sensitization”, and was previously found only in embryonic cells. Since further elucidation of the molecular mechanisms that are responsible for such sensitization in primary adMSCs was difficult due to the high heterogeneity in adrenergic receptor expression, we employed immortalized adipose-derived mesenchymal stem cell lines (hTERT-MSCs). Using flow cytometry and immunofluorescence microscopy, we demonstrated that the proportion of cells expressing adrenergic receptor isoforms does not differ significantly in hTERT-MSCs cells compared to the primary adMSCs culture. However, using analysis of Ca^2+^-mobilization in single cells, we found that these cells did not demonstrate the sensitization seen in primary adMSCs. Consistently, these cells did not activate cAMP synthesis in response to noradrenaline. These data indicate that immortalized adipose-derived mesenchymal stem cell lines demonstrated impaired ability to respond to noradrenaline compared to primary adMSCs. These data draw attention to the usage of immortalized cells for MSCs-based regenerative medicine, especially in the field of pharmacology.

## 1. Introduction

Adipose tissue renewal, as well as its growth during high-fat diet, strongly depends on the activity of progenitor cells, residing in adipose tissue stroma, known under various names, including mesenchymal stem/stromal cells (MSCs) or adipose derived stromal cells, etc. [1,2]. Hormones and neurotransmitters could direct the functional activity of these cells [3,4]; however, the precise mechanisms of regulation involved are not clear. 

Previously, we demonstrated that the ratio of adipose derived MSCs (adMSCs) subpopulations expressing α1A, α1B, α2A, α2B, β1, β2, and β3 adrenergic receptors vary substantially from donor to donor. Besides, not every cell which expresses a particular receptor was able to respond to the appropriate hormone [5]. adMSCs may be divided into the functional subpopulations according to their ability to respond to the particular hormone [6]. Primary adMSCs may shift their functional activity by changing their sensitivity to particular hormones. We demonstrated an unexpected mechanism of switching of intracellular signaling of adMSCs from beta-adrenoceptor/cAMP to alpha1A-adrenoceptor/calcium-dependent pathways, which caused an increase in sensitivity to catecholamines [5,7]. That mechanism termed “heterologous sensitization” may be used by cells for moving from one functional subpopulation to another. 

Further investigation of signaling mechanisms of adrenergic regulation of adMSCs required the population with more homogenous expression of adrenoceptors. In this study, we used a commercially-available cell line of immortalized adipose tissue-derived MSCs (hTERT-MSCs) with defined characteristics [8]. Although these cells demonstrated rather stable expression of adrenoceptors and responsivity to noradrenaline, hTERT-MSCs ability to regulate their sensitivity to noradrenaline was impaired. These data draw attention to the usage of immortalized cells for MSCs-based regenerative medicine, especially in the field of pharmacology.

## 2. Results

### 2.1. Analysis of Adrenergic Receptors Expression in hTERT-MSCs

To identify adrenergic receptor isoforms expressed by hTERT-MSCs, we performed immunofluorescent staining using specific antibodies followed by confocal microscopy analysis. hTERT-MSCs expressed α1A, α1B, α2A, α2B, β1, β2, and β3 adrenergic receptors at their surface and intracellular, with the pattern similar to primary adMSCs (Figure 1). Flow cytometry analysis demonstrated that the percentage of the cells containing particular adrenergic receptor isoforms was rather stable, but lower compared to primary cultures (Figure 2). Similar to primary cells, α-adrenergic receptors, in particular, α1A, were the most abundant in hTERT-MSCs, albeit at much lower level, as relative MFI for α1A was 3.5 times higher in adMSCs (Figure 2). At the same time, hTERT-MSCs contained low proportions of cells expressing α1B, α2A, α1B, and β1 isoforms (Figure 2).

### 2.2. Ca^2+^ Signaling in Single Cells

As we showed in our previous studies, primary adMSCs demonstrate high heterogeneity at the functional level, i.e., in noradrenaline sensitivity [5,6]. Thus, 7.5 ± 0.8% of primary adMSCs responded to serial noradrenaline applications by Ca^2+^ release. As shown in Table 1, primary adMSCs demonstrated high variation in noradrenaline responsiveness, depending on the particular donor (Figure 3A). We supposed that hTERT-MSCs, being a cell line, are less heterogeneous and would respond to the hormone uniformly. However, using the registration of intracellular Ca^2+^ signaling in single cells, we demonstrated that hTERT-MSCs respond to noradrenaline in a non-uniform manner; only 1.9 ± 0.3% of these cells responded to noradrenaline by calcium release (Figure 3A–С). Variation of hTERT-MSCs noradrenaline responsiveness calculated as relative SD (RSD) was 75%, which is comparable to primary adMSCs. Thus, immortalized hTERT-MSC did not respond to noradrenaline uniformly. The percentage of noradrenaline responding cells was 4 times lower compared to primary adMSCs, and a responsiveness variation of hTERT-MSCs was similar to primary cells.

Previously, we showed that primary adMSCs respond to noradrenaline with Ca^2+^ release using both α1- and α2-adrenoceptors [5]. Here, we analyzed which isoforms of adrenoceptors are responsible for noradrenaline-dependent Ca^2+^ increase in hTERT-MSCs. We stimulated hTERT-MSCs, as well as primary adMSCs, with either α1-agonist phenylephrine or α2-agonist clonidine, and measured the number of responding cells. We showed that the share of hTERT-MSCs which responded to phenylephrine was comparable to that of noradrenaline (Figure 3D), and significantly lower than in primary adMSCs. The share of hTERT-MSCs which responded to clonidine was close to zero (Figure 3E). Thus, hTERT-MSCs responded to noradrenaline in a calcium-dependent manner predominantly using α1-adrenoceptors, but at a significantly lower level compared to primary adMSCs.

### 2.3. Heterologous Sensitization in hTERT-MSCs 

Next, we analyzed whether heterologous sensitization phenomena is present in hTERT-MSCs cells. We stimulated hTERT-MSCs, as well as primary adMSCs, with 10^–6^ M noradrenaline for 1 h and analyzed their response to the fresh addition of this hormone 6 h later. Consistent with our previous observation, in primary adMSCs, noradrenaline caused more than a 2-fold increase of proportion of cells capable of responding to this hormone by the activation of Ca^2+^ signaling (Figure 4A,C). However, noradrenaline did not cause an increase of the ratio of noradrenaline responding cells in hTERT-MSCs (Figure 4A,B). Thus, using analysis of calcium signaling in single cells, we demonstrated that the immortalized adipose-derived mesenchymal stem cell line shows impaired ability to regulate noradrenaline sensitivity compared to primary adMSCs. 

### 2.4. cAMP Signaling

As we have shown, heterologous sensitization of primary adMSCs was dependent on the activation of beta-adrenoceptors and cAMP synthesis [5]. Here, we examined whether noradrenaline could activate cAMP synthesis in hTERT-MSCs. Using an ELISA-based method, we showed that noradrenaline stimulated cAMP synthesis in primary adMSCs, whereas in hTERT-MSCs, it did not (Figure 4D,E). Thus, hTERT-MSCs were not able to activate cAMP synthesis in response to noradrenaline, despite of the presence of beta-adrenergic receptors. Furthermore, forskolin also failed to activate cAMP synthesis in hTERT-MSCs, indicating impaired adenylate cyclase expression or activity in these cells. Such disabled cAMP activation could be responsible for the impaired regulation of noradrenaline sensitivity in hTERT-MSCs. 

## 3. Discussion

Intrinsic heterogeneity, prominent donor-to-donor variation, and high tolerance of primary adMSCs to common transfection methods, as well as the need for a well-characterized cell line for cell therapy, led to the establishment of the hTERT immortalized adipose derived mesenchymal stem cell line [8]. To date, these cells were used in a variety of basic studies, including those focused on the regulatory mechanisms of YAP-dependent mechanosensing, Nanog-mediated pluripotency maintenance, and others [9,10,11]. These studies encouraged us to employ hTERT-MSCs cells to dissect the molecular mechanisms underlying the intriguing ability of primary adMSCs to increase their sensitivity to noradrenaline in response to this hormone [5]. The immortalized adipose derived mesenchymal cell line ASC52Telo, referred to here as “hTERT-MSCs”, was produced using a retroviral transduction for introduction of hTERT and G418 selection [8]. Due to the clonal selection of the fastest-growing cells in long-cultivating population, we supposed that hTERT-MSC respond more uniformly to noradrenaline action. However, the expression of α and β adrenergic receptors was rather low in these cells. Furthermore, by measuring intracellular calcium signaling at the single cell level, we showed that this cell line retained functional heterogeneity that was similar to primary adMSCs. Only a small part of hTERT-MSCs population responded to noradrenaline with calcium influx; a variability of calcium responses was also similar to primary adMSCs. In contrast to primary cells, hTERT-MSCs did not respond to α2-agonist clonidine, which corresponds to the low expression level of α2-adrenoceptors in these cells.

Most importantly, in contrast to primary cells, hTERT-MSCs failed to up-regulate their sensitivity to noradrenaline in response to this hormone. Previously, we demonstrated that in primary MSCs, such heterologous sensitization depended at the activation of beta-adrenergic receptors and cAMP synthesis. In hTERT-MSCs, noradrenaline did not stimulate the activation of cAMP synthesis, despite of the presence of beta-adrenergic receptors. 

Recently, a single-cell transcriptomic approach has revealed several subpopulations within primary adMSCs of adipose tissue [12]. Among those, a regulatory subpopulation that is responsible for the overall adipogenic differentiation response was identified. Our data indicate that due to the methodological peculiarities (i.e., clonal selection after viral transduction or different susceptibility of functional subpopulations to retroviral transduction), the hTERT-MSC cell line could be deprived of some subpopulations that may regulate the noradrenaline sensitivity of the rest of the cells.

hTERT-MSCs have a great potential as a source of cells for various cell therapeutic approaches, since they retain differentiation capabilities similar to primary adMSCs, and could be amplified in desired quantities. Such immortalized cells were already used for bone repair and in an implantable glucose-sensing device [13,14]. However, our data indicate that hormonal regulation of these cells differs from primary adMSCs. This should be taken into consideration for the usage of the hTERT-MSCs cell line in applications assuming a correct hormonal control of implanted cells in situ. These data also draw attention to the usage of immortalized cells for MSCs-based regenerative medicine, especially in the field of pharmacology.

## 4. Materials and Methods 

### 4.1. Cell Culture

ASC52telo, hTERT immortalized adipose derived mesenchymal stem cells (ATCC® SCRC-4000™) were obtained from ATCC (Manassas, VA, USA) and maintained in Advance Mesenchymal Cell Medium (HyClone, GE Healthcare, Chicago, IL, USA) in a humidified CO_2_ incubator with 5% CO_2_ and 95% air, 37 °C. Cells were passaged regularly at 70–80% of confluency. Human primary adMSCs from subcutaneous fat tissue of healthy young donors were obtained from biobank of Lomonosov Moscow State University [15]. All donors gave their informed consent, and the local ethics committee of Lomonosov Moscow State University Medical Research and Education Center (Moscow, Russia) approved the study protocol (Study protocol #4, 4 June 2018). Cell identity, including expression of specific surface markers and differentiation capabilities, were confirmed by storage facility. Cells were cultured in AdvanceSTEM Mesenchymal Stem Cell Media containing 10% AdvanceSTEM Supplement (HyClone), 1% antibiotic-antimycotic solution (HyClone) at 37 °C in 5% CO_2_ incubator. Cells were passaged at 70% confluency using HyQTase solution (HyClone). For the experiments, adMSCs cultured up to 3rd- 4th passages were used.

### 4.2. Flow Cytometry

The ratio of cells expressing particular adrenergic receptor was examined using flow cytometry as previously described. Briefly, adrenergic receptors were detected using specific antibodies α1A (abcam, Cambridge, UK ab137123, dilution 1:250), α1B (abcam ab169523, 1:100), α2A (abcam ab65833, 1:100), α2B (abcam ab151727, 1:100), β1 (Thermo Fisher Scientific, Waltham, MA, USA, PA1-049, 1:100), β2 (Thermo Fisher Scientific PA5-19649, 1:100), β3(Abnova, Taipei, Taiwan H00000155-B01P, 1:100), following by secondary antibodies AlexaFluor 594 (Invitrogen, Waltham, MA, USA, 1:500) on formalin fixed non permeabilized cells. Normal rabbit IgG (BD Pharmingen, Franklin Lakes, NJ, USA, 1:300) and normal mouse IgG1 (BD Pharmingen, 1:400) were used as a negative control. Stained cells were analyzed using flow cytometry scanner BDLSR Fortessa Special Order Research Product (BD Pharmingen, USA). Finally, 20,000–40,000 events were acquired and analyzed for antigen expression. 

### 4.3. Immunofluorescence

Adrenergic receptors were stained using specific antibodies α1A (abcam ab137123, dilution 1:100), α1B (abcam ab169523, 1:100), α2A (abcam ab65833, 1:100), α2B (abcam ab151727, 1:100), β1 (Thermo Fisher Scientific PA1-049, 1:100), β2 (Thermo Fisher Scientific PA5-19649, 1:100), β3 (Abnova, H00000155-B01P, 1:100), following by secondary antibodies AlexaFluor 594 (Invitrogen, 1:500) on formalin fixed non permeabilized cells. Normal rabbit IgG (BD Pharmingen, 1:300) and normal mouse IgG1 (BD Pharmingen, 1:400) were used as a negative control. The presence of adrenergic receptors was analyzed using immunofluorescent staining, and analyzed using a Leica DMI6000D (Leica Microsystems, Wetzlar, Germany) microscope equipped with a camera Leica DFC 7000T.

### 4.4. Ca^2+^ Signaling

Adrenergic receptor activation was assessed using noradrenaline and Ca^2+^ imaging at the single cell level. Cells grown in HyClone Advance Stem medium with a Supplement in 24 well plates were loaded with Fluo-8 (abcam, ab142773), 4 µM in Hanks Balanced Salt Solution with 20 mM Hepes, for 1 h. Cells were grown at low densities to prevent cell-to-cell communications during the calcium imaging. To analyze the functional activity, cells were treated with either noradrenaline (Calbiochem, Darmstadt, Germany, Cat# 489350, 1 µM), or α1-agonist phenylephrine (Abcam, ab120761, 100 μM) or α2-adrenergic receptor agonist clonidine (Abcam, ab120753, 100 μM). To analyze the amount of functionally active cells after pretreatment with noradrenaline, cells were stimulated with noradrenaline for 1 h, washed three times, and then incubated in full growth medium for an additional 5 h. To measure the percent of responding cells, we recorded the baseline for 5 min, and then once added noradrenaline. Ca^2+^ transients were measured in individual cells using an inverted fluorescent microscope Nikon Eclipse Ti equipped with an objective CFI Plan Fluor DLL 10X/0.3 (Nikon, Tokyo, Japan) and with digital EMCCD camera Andor iXon 897 (Andor Technology, Belfast, UK). We used the simultaneous measuring of 6×6 fields of view in Large Image mode to increase the number of analyzed cells. Movies were analyzed using NIS-Elements (Nikon) and ImageJ software (NIH, USA). Alterations of cytosolic Ca^2+^ from the resting level were quantified by a relative change in the intensity of Fluo-8 fluorescence (ΔF/F_0_) recorded from an individual cell. The percent of responding cells was measured as a ratio of the number of responding cells to the number of all analyzed cells.

### 4.5. cAMP ELISA

To measure cAMP-dependent response of the cells, we used cAMP Direct Fluorometric Immunoassays Kit (abcam, ab138880). Cells were grown in a 96-well plate to a near confluent monolayer a day before the experiment. Cells were treated with noradrenaline for 15 min and lysed with Cell Lysis Buffer for 10 min. Samples were prepared, and a cAMP Assay procedure was performed according to the manufacturer’s protocol. cAMP concentrations in the samples were calculated using a calibration curve preparing in every experiment independently. 

### 4.6. Statistics

Statistical analysis was performed using SigmaPlot 11.0 software (Systat Software Inc., San Jose, CA, USA). Data was assessed for normality of distribution using the Kolmogorov-Smirnov test. Values are expressed as mean ± standard error of mean (SEM). Comparison of two independent groups was performed by Student *t*-test for normally-distributed data, and Mann–Whitney U-criteria (M-U test) for not normally distributed data. Variation of calcium responses were measured for normally-distributed samples as relative standard deviations (RSDs), calculated as standard deviations divided by the mean value. Statistical significance was defined as *p*-value < 0.05. 

## Figures and Tables

**Figure 1 ijms-19-03712-f001:**
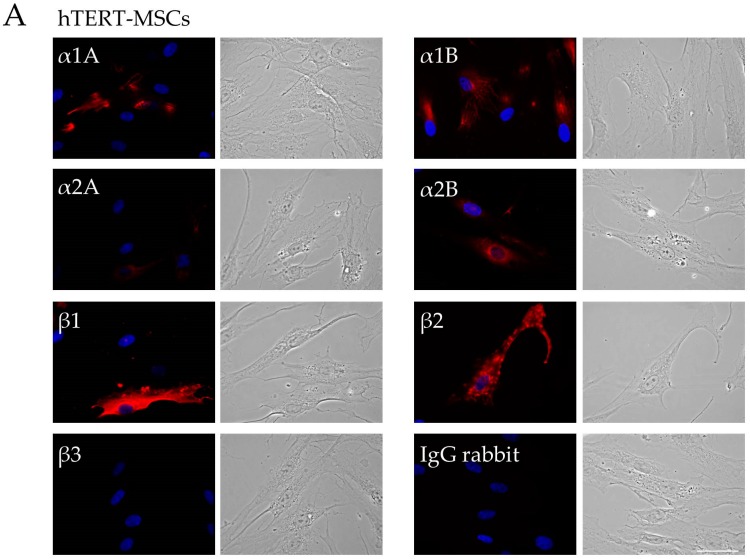
Immunofluorescent staining of adrenergic receptors in hTERT-MSCs (**A**) and primary adMSCs (**B**). Representative images of immunofluorescent staining of particular adrenergic receptors isoforms and corresponding bright field images. Receptor isoform name is marked in upper-left corner of each immunofluorescent image. Mouse IgG controls are the same as rabbit IgG and are not shown. Scale bar 50 μm.

**Figure 2 ijms-19-03712-f002:**
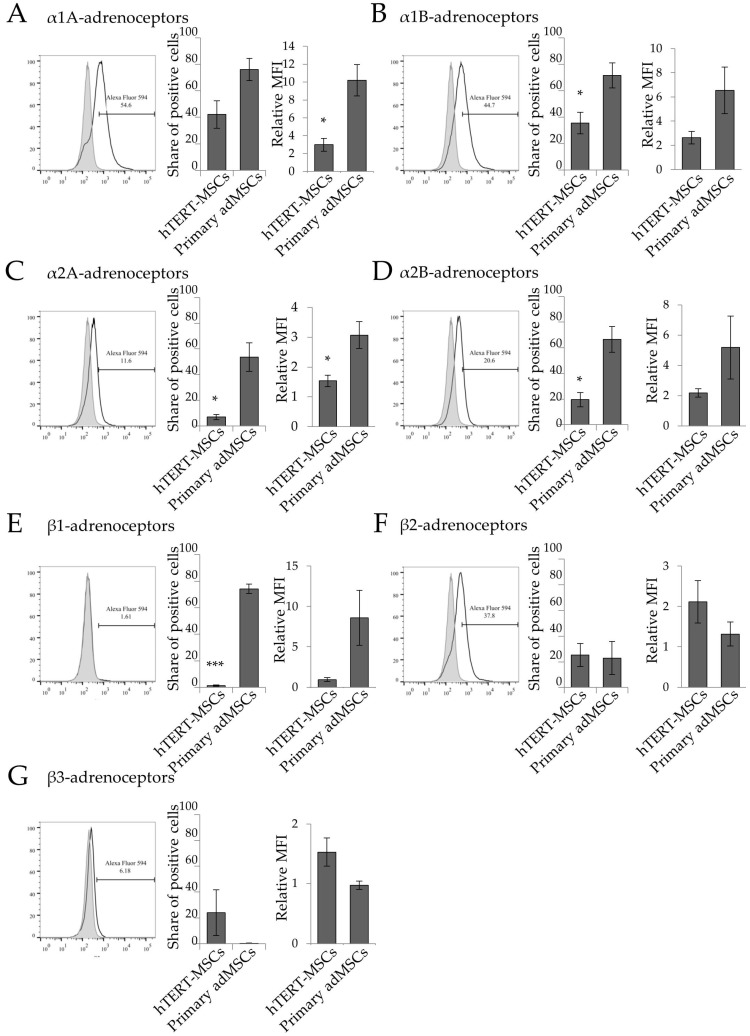
Flow cytometry analysis of adrenergic receptors expression at the hTERT-MSCs surface. Representative flow cytometry histograms and comparative statistical analysis of α1A- (**A**), α1B- (**B**), α2A- (**C**), α2B- (**D**), β1- (**E**), β2- (**F**), and β3-adrenoceptors (**G**) expression in hTERT-MSCs and primary MSCs are shown [7]. Comparison of share of cells expressing particular adrenergic receptors are shown at the left diagrams in every section of figure. Comparison of relative median fluorescence intensity (MFI) are shown at the right diagrams in the everu section. Mean ± SEM, comparisons of share of positive cells and relative MFI of α1B-, and α2B-isoforms were performed using Mann–Whitney U-criteria (M-U test) because of abnormal distribution of the data. Comparisons of other data were performed by Student *t*-test. *n* = 3 for hTERT-MSCs and *n* = 4–9 for primary adMSCs. * *p* < 0.05, *** *p* < 0.001.

**Figure 3 ijms-19-03712-f003:**
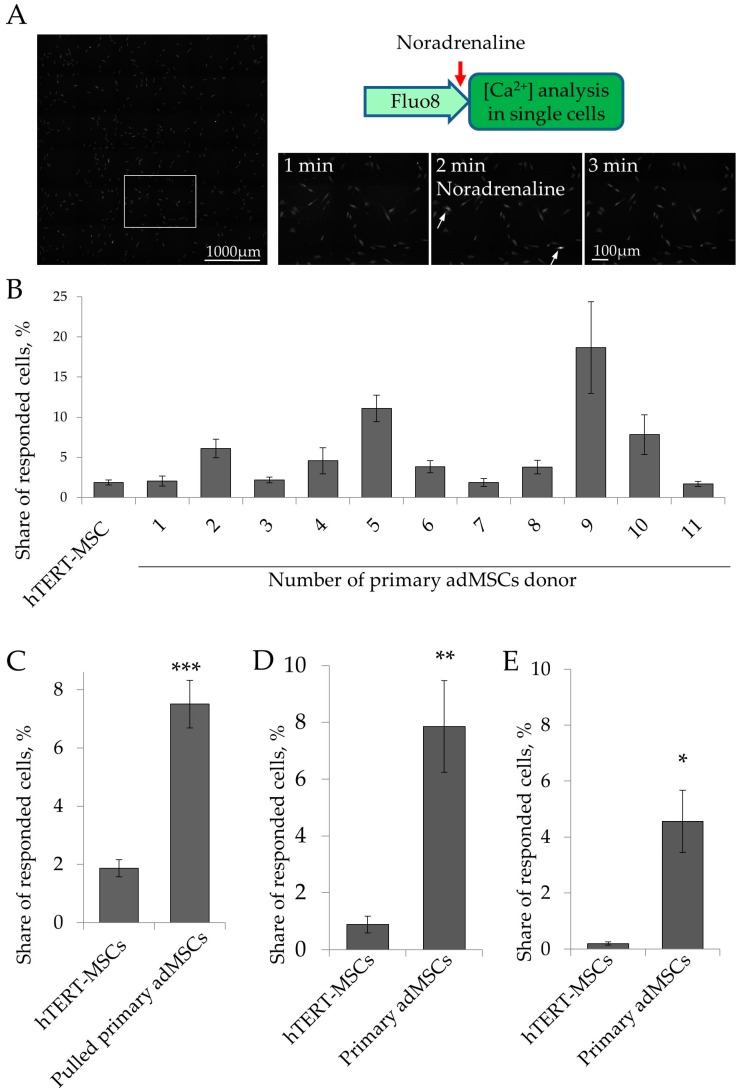
hTERT-MSCs demonstrated impaired sensitivity to noradrenaline. (**A**) Scheme of experiment and representative images of Ca^2+^ registration in single cells using Fluo-8 dye. Large field of view and 100–200 cells were analyzed simultaneously. (**B**) Share of hTERT-MSCs and primary adMSCs, derived from different donors and that responded to noradrenaline applications by Ca^2+^ release. (**C**) Share of pulled sample of primary adMSCs that responded to noradrenaline applications by Ca^2+^ release compared to hTERT-MSCs. (**D**) Share of hTERT-MSCs and primary adMSCs that responded to α1-agonist phenylephrine (10^−4^ M) applications by Ca^2+^ release. (**E**) Share of hTERT-MSCs and primary adMSCs that responded to α2-agonist clonidine (10^−4^ M) applications by Ca^2+^ release. Mean ± SEM, comparison was performed by Mann–Whitney U-criteria (M-U test) because of not normally distribution of the data, *n* = 4–22 for (**B**), *n* = 22–139 for (**C**), *n* = 5–16 for (**D**), *n* = 5–9 for (**E**). * *p* < 0.05, ** *p* < 0.01, *** *p* < 0.001.

**Figure 4 ijms-19-03712-f004:**
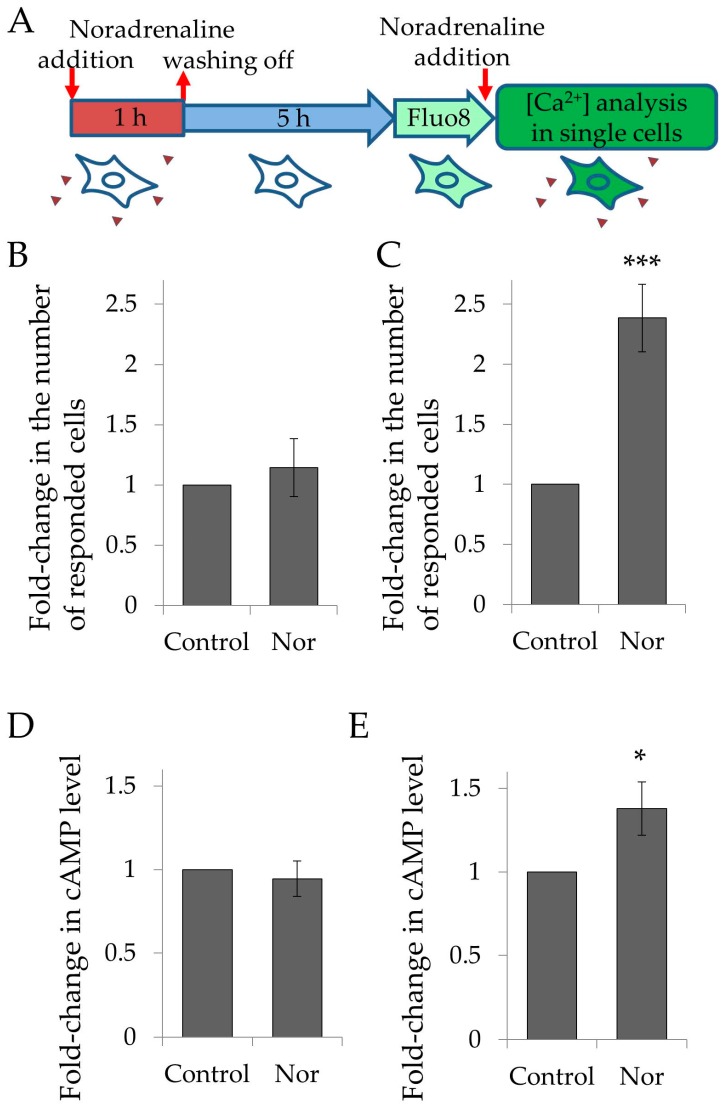
hTERT-MSCs demonstrated impaired regulatory mechanisms of noradrenaline-dependent intracellular signaling. (**A**) Scheme of experiments measuring heterologous sensitization in adMSCs. (**B**) and (**C**) Heterologous sensitization in hTERT-MSCs and in primary adMSCs, respectively. Cells were treated with noradrenaline for 1 h and responsiveness was measured 6 h later. Control-cells were treated with growth medium; Nor-cells were treated with 10^−6^ M noradrenaline. (**D**) and (**E**) Noradrenaline-induced changes in cAMP level measured ELISA-based assay in hTERT-MSCs (**D**) and in primary adMSCs (**E**). Control-cells were treated with growth medium; Nor-cells were treated with 10^−6^ M noradrenaline for 15 min. Mean ± SEM, comparison was performed by Mann–Whitney U-criteria (M-U test) because of not normally distribution of the data, *n* = 12 for (**B**), *n* = 8 for (**C**), *n* = 8 for (**D**), *n* = 4 for (**E**). * *p* < 0.05, *** *p* < 0.001.

**Table 1 ijms-19-03712-t001:** Variation in Responsivity of Primary adMSCs.

Primary adMSCs Donor ID	Mean of Responding Cells, %	SD	Relative SD (RSD), %
1	2.03	1.56	76
2	6.1	2.34	38
3	2.18	0.87	39
4	4.56	4.56	100
5	11.08	7.54	68
6	3.83	2.58	67
7	1.86	1.63	87
8	3.78	1.91	50
9	18.68	15.06	80
10	7.82	6.52	83
11	1.7	0.97	57
		**Mean RSD**	68

SD is standard deviation, RSD is relative standard deviation.

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
