# Peer review of "Noradrenaline Sensitivity Is Severely Impaired in Immortalized Adipose-Derived Mesenchymal Stem Cell Line"

_ijms, 2018, doi:10.3390/ijms19123712_

Round 1

Reviewer 1 Report

The manuscript entitled “Noradrenaline sensitivity is severely impaired in immortalized adipose-derived mesenchymal stem cell line” by Pyotr A. Tyurin-Kuzmin and colleagues addresses the question of transferability of data obtained from primary adipose tissue-derived mesenchymal stem/stromal cells (adMSCs) to that obtained using immortalized adipose tissue-derived MSCs (hTERT-MSCs). To compare both data sets in terms of transferability, they assessed (i) the expression of adrenergic receptor (AR) isoforms using immunofluorescence microscopy and flow cytometry. Furthermore, the authors analyzed (ii) the sensitivity of these receptors using the catecholamine noradrenaline by Ca2+ imaging at the single cell level using inverted fluorescent microscopy and by cAMP-dependent response of the cells using cAMP Direct Fluorometric Immunoassays Kit.

The investigated topic is timely and important for the understanding of divergences in the population of primary adMSCs and the standardized use of immortalized cells such as hTERT-MSCs especially in the field of pharmacology and pharmacodynamics. Although I highly appreciate the research topic, I have some major and minor concerns with regard to the data acquisition, analysis, presentation and discussion. In general, it would be helpful for the reader to provide a clear aim and/or hypothesis.

Major points:

Comparing freshly acquired data using flow cytometry with data acquired in previous studies may provide different results due to aging of lasers, varying calibrations, loss of antibody conjugation quality and different staining procedures. Thus, I highly recommend the provision of a standardized control staining using the same sample over the time-period analyzed in order to monitor changes of the instrument and the experimenter.

In cases where the entire population stains (the whole population shifts!) with different levels of an antibody – like measuring expression levels of ARs – I recommend to measure the shift in fluorescence intensity (MFI, but use median instead of mean!) of the population of cells. It is recommended to report relative MFI values based on some sort of control (unstained, isotype, FMO, etc...) to demonstrate an increase or decrease in expression of this marker (assuming that each sample was stained with saturating amounts of antibody, and all samples were run under the same conditions and instrument settings – see point above). Thus, please provide these data.

Focusing on the IF data, I recommend to present brightfield images, higher magnifications and IF data from primary adMSCs, in order to provide insights into the distribution of AR expression (ic or surface) and a direct comparison to the expression in primary adMSCs (see figure 1).

For comparison of hTERT-MSCs with adMSCs with regard to AR expression by flow cytometry (see figure 2), histograms should be provided as overlays e.g. isotype control, staining of hTERT-MSCs and staining of adMSCs. The authors have to provide relative MFIs of each, which they have to present as scatter dot plot.

Previously, the authors demonstrated that the α2A-isoform coupled to PLC endowed MSCs with sensitivity to noradrenaline. Is this the case also for hTERT-MSCs? This should be tested using inhibitors, to provide functional evidence.

Minor points:

With regard to Ca2+-signaling the authors noted that they took movies from cells flashing up with fluo-3AM optically demonstrating Ca2+-signaling. Representative figures (in series) should be provided for the reader demonstrating (a) the method and (b) the areas of detection.

Previously the authors demonstrated that the α2A-isoform coupled to PLC endowed MSCs with sensitivity to noradrenaline. Is this the case also for hTERT-MSCs? This should be tested using inhibitors, to provide functional evidence.

Statistics are not properly indicated and have to be revised for each figure.

I recommend a gating strategy for both hTERT-MSCs and adMSCs.

Discussion: Previously, the authors demonstrated that the α2A-isoform coupled to PLC endowed MSCs with sensitivity to noradrenaline. Is this the case also for hTERT-MSCs?

Author Response

Responses to Reviewers

Manuscript ID: ijms-385477

Type of manuscript: Communication

Title: Noradrenaline sensitivity is severely impaired in immortalized adipose-derived mesenchymal stem cell line

Corresponding Author: Dr. Pyotr A. Tyurin-Kuzmin, Natalia I. Kalinina
Authors: Pyotr A. Tyurin-Kuzmin, Vadim I. Chechekhin, Anastasiya M. Ivanova, Daniyar T. Dyikanov, Veronika Y. Sysoeva, Natalia I. Kalinina, Vsevolod A. Tkachuk

On behalf of the authors team I would like to thank Reviewers of our manuscript entitled “Noradrenaline sensitivity is severely impaired in immortalized adipose-derived mesenchymal stem cell line” for their valuable comments, which helped us to improve our manuscript. We have made changes in the manuscript text, as well as performed additional experiments and data analysis, which were required by reviewers. All changes in the manuscript body are highlighted by yellow color. I believe that these changes have strengthened our manuscript and made it consistent with the overall quality of IJMS.

Responses to Reviewer #1:

Major points:

1. Comparing freshly acquired data using flow cytometry with data acquired in previous studies may provide different results due to aging of lasers, varying calibrations, loss of antibody conjugation quality and different staining procedures. Thus, I highly recommend the provision of a standardized control staining using the same sample over the time-period analyzed in order to monitor changes of the instrument and the experimenter.

Naturally, fluorophore as well as device properties do change over time. Therefore, internal IgG control was included in each independent experiment and all measurements (positive cell ratio or median fluorescence intensity) were normalized to the control.

2. In cases where the entire population stains (the whole population shifts!) with different levels of an antibody – like measuring expression levels of ARs – I recommend to measure the shift in fluorescence intensity (MFI, but use median instead of mean!) of the population of cells. It is recommended to report relative MFI values based on some sort of control (unstained, isotype, FMO, etc...) to demonstrate an increase or decrease in expression of this marker (assuming that each sample was stained with saturating amounts of antibody, and all samples were run under the same conditions and instrument settings – see point above). Thus, please provide these data.

We now reported relative MFI values, data included in Figure 2 of revised manuscript. We also refer to relative MFI values in the text of revised manuscript.

3. Focusing on the IF data, I recommend to present brightfield images, higher magnifications and IF data from primary adMSCs, in order to provide insights into the distribution of AR expression (ic or surface) and a direct comparison to the expression in primary adMSCs (see figure 1).

We now supplied 63x magnification images of adMSCs and TERT-MSCs stained with antibodies against adrenoceptors. We also added brightfield images corresponded to all IF pictures (please, see Figure 1 of revised manuscript).

4. For comparison of hTERT-MSCs with adMSCs with regard to AR expression by flow cytometry (see figure 2), histograms should be provided as overlays e.g. isotype control, staining of hTERT-MSCs and staining of adMSCs. The authors have to provide relative MFIs of each, which they have to present as scatter dot plot.

We now reported relative MFI values, data included in Figure 2 of revised manuscript. As for overlayed histograms, we created those but found such data presentation confusing (please, see Supplementary figure 1 below). Therefore, we decided to include in revised Figure 2 only representative histograms demonstrating receptors expression in hTERT-MSC and relative MFI we now presented as additional graphs on this figure. If Reviewer finds appropriate – we are willing to include overlayed histograms as a supplementary material.

5. Previously, the authors demonstrated that the α2A-isoform coupled to PLC endowed MSCs with sensitivity to noradrenaline. Is this the case also for hTERT-MSCs? This should be tested using inhibitors, to provide functional evidence.

We performed additional experiments with α1-agonist phenylephrine and α2-agonist clonidine to test which isoforms of adrenoceptors are responsible for noradrenaline-dependent stimulation of calcium signaling in hTERT-MSC. Using this approach, we confirmed that hTERT-MSCs respond by Ca2+ influx only by α1-adrenoreceptors. These data are now included in revised manuscript.

Minor points:

1. With regard to Ca2+-signaling the authors noted that they took movies from cells flashing up with fluo-3AM optically demonstrating Ca2+-signaling. Representative figures (in series) should be provided for the reader demonstrating (a) the method and (b) the areas of detection.

We added appropriate schemes of experiments in Fig 3 and Fig 4, as wells as representative figures in series in revised Figure 3.

2. Previously the authors demonstrated that the α2A-isoform coupled to PLC endowed MSCs with sensitivity to noradrenaline. Is this the case also for hTERT-MSCs? This should be tested using inhibitors, to provide functional evidence.

We performed additional experiments with α1-agonist phenylephrine and α2-agonist clonidine to test which isoforms of adrenoceptors are responsible for noradrenaline-dependent stimulation of calcium signaling in hTERT-MSC. Using this approach, we confirmed that hTERT-MSCs respond by Ca2+ influx only by α1-adrenoreceptors. These data are now included in revised manuscript.

3. Statistics are not properly indicated and have to be revised for each figure.

We now included statistics description in each figure in the revised manuscript.

4. I recommend a gating strategy for both hTERT-MSCs and adMSCs.

The same gating strategy was used for both hTERT-MSCs and adMSC.

5. Discussion: Previously, the authors demonstrated that the α2A-isoform coupled to PLC endowed MSCs with sensitivity to noradrenaline. Is this the case also for hTERT-MSCs?

We added this point to Discussion section (please, see revised manuscript for details).

Reviewer 2 Report

General Comments

Need a stronger statement indicating the purpose of the study.  Furthermore, need a concise statement of the biological/medical significance of the work.  Suggest revising Abstract, Introduction, Discussion to include this information.

Technical work appears to be adequate.

Text is well written in English.

Minor problems with punctuation.

Specific Comments

Introduction, l,34: It is always important to better define MSCs by including their source, e.g., bone marrow, adipose tissue, etc.

 Author Response

Responses to Reviewers

Manuscript ID: ijms-385477

Type of manuscript: Communication

Title: Noradrenaline sensitivity is severely impaired in immortalized adipose-derived mesenchymal stem cell line

Corresponding Author: Dr. Pyotr A. Tyurin-Kuzmin, Natalia I. Kalinina
Authors: Pyotr A. Tyurin-Kuzmin, Vadim I. Chechekhin, Anastasiya M. Ivanova, Daniyar T. Dyikanov, Veronika Y. Sysoeva, Natalia I. Kalinina, Vsevolod A. Tkachuk

On behalf of the authors team I would like to thank Reviewers of our manuscript entitled “Noradrenaline sensitivity is severely impaired in immortalized adipose-derived mesenchymal stem cell line” for their valuable comments, which helped us to improve our manuscript. We have made changes in the manuscript text, as well as performed additional experiments and data analysis, which were required by reviewers. All changes in the manuscript body are highlighted by yellow color. I believe that these changes have strengthened our manuscript and made it consistent with the overall quality of IJMS.

Responses to Reviewer:

General Comments

1. Need a stronger statement indicating the purpose of the study.  Furthermore, need a concise statement of the biological/medical significance of the work.  Suggest revising Abstract, Introduction, Discussion to include this information.

We have added concise statement of biological/medical significance of this study in Abstract, Introduction, and Discussion sections.

2. Technical work appears to be adequate.

3. Text is well written in English.

4. Minor problems with punctuation.

We have revised manuscript and corrected the punctuation in some places

Specific Comments

Introduction, l,34: It is always important to better define MSCs by including their source, e.g., bone marrow, adipose tissue, etc.

We have changed abbreviation ‘MSCs’ to ‘adMSC’, as 'adipose derived MSCs'.

Round 2

Reviewer 1 Report

The authors have addressed all concerns raised from the first version of the manuscript. I agree with the authors to provide the histogram overlays as supplementary figure 1. The manuscript is now suitable for publication. Well done!